

# Questions of time and affect: a person's affectivity profile, time perspective, and well-being

Danilo Garcia[1,2,3,4,5], Uta Sailer[2,3], Ali Al Nima[1,3] and Trevor Archer[2,3]

[1] Blekinge Center of Competence, Blekinge County Council, Karlskrona, Sweden
[2] Department of Psychology, University of Gothenburg, Gothenburg, Sweden
[3] Network for Empowerment and Well-Being, University of Gothenburg, Gothenburg, Sweden
[4] Institute of Neuroscience and Physiology, Sahlgrenska Academy, Gothenburg, Sweden
[5] Department of Psychology, Lund University, Lund, Sweden

Corresponding author
Danilo Garcia,
danilo.garcia@icloud.com

## ABSTRACT

**Background.** A "balanced" time perspective has been suggested to have a positive influence on well-being: a sentimental and positive view of the past (high Past Positive), a less pessimistic attitude toward the past (low Past Negative), the desire of experiencing pleasure with slight concern for future consequences (high Present Hedonistic), a less fatalistic and hopeless view of the future (low Present Fatalistic), and the ability to find reward in achieving specific long-term goals (high Future). We used the affective profiles model (i.e., combinations of individuals' experience of high/low positive/negative affectivity) to investigate differences between individuals in time perspective dimensions and to investigate if the influence of time perspective dimensions on well-being was moderated by the individual's type of profile.

**Method.** Participants ($N = 720$) answered to the Positive Affect Negative Affect Schedule, the Zimbardo Time Perspective Inventory and two measures of well-being: the Temporal Satisfaction with Life Scale and Ryff's Scales of Psychological Well-Being-short version. A Multivariate Analysis of Variance (MANOVA) was conducted to identify differences in time perspective dimensions and well-being among individuals with distinct affective profiles. Four structural equation models (SEM) were used to investigate which time perspective dimensions predicted well-being for individuals in each profile.

**Results.** Comparisons between individuals at the extreme of the affective profiles model suggested that individuals with a self-fulfilling profile (high positive/low negative affect) were characterized by a "balanced" time perspective and higher well-being compared to individuals with a self-destructive profile (low positive/high negative affect). However, a different pattern emerged when individuals who differed in one affect dimension but matched in the other were compared to each other. For instance, decreases in the past negative time perspective dimension lead to high positive affect when negative affect is high (i.e., self-destructive vs. high affective) but to low negative affect when positive affect was high (i.e., high affective vs. self-fulfilling). The moderation analyses showed, for example, that for individuals with a self-destructive profile, psychological well-being was significantly predicted by the past negative, present fatalistic and future time perspectives. Among individuals with a high affective or a self-fulfilling profile, psychological well-being was significantly predicted by the present fatalistic dimension.

**Conclusions.** The interactions found here go beyond the postulation of a "balanced" time perspective being the only way to promote well-being. Instead, we present a more person-centered approach to achieve higher levels of emotional, cognitive, and psychological well-being.

"*When I was a boy, what joy*

*playing war night and day*

*jumping over a fence to see you*

*and like that discover something new in your eyes*"

*From the song "El gato que esta triste y azul" [The Cat Who Is Sad and Blue] performed by Roberto Carlos and written by Giancarlo Bigazzi and Toto Savio.*

According to *Zimbardo & Boyd* (*1999*), time perspective is the process of assigning experiences "to temporal categories, or time frames, that help to give order, coherence, and meaning to those events." The mental organization of time is typically anchored in the time referents of past, present, and future (*Shmotkin & Eyal, 2003*). The way in which an individual evaluates each of these constitutes her/his time perspective, or time orientation (*Wallace & Rabin, 1960*). According to time perspective theory, the way individuals view their past, present, and future influences their decisions and behavior.

One of the most widely-used measures of time perspective is the Zimbardo Time Perspective Inventory (ZTPI; *Zimbardo & Boyd, 1999*). The ZTPI measures five time dimensions: (1) *Past-Positive*, a nostalgic, positive attitude towards the past that is positively related to high self-esteem and happiness; (2) *Past-Negative*, a generally negative view of the past that is positively related to depression, low self-esteem, anxiety, sadness and aggression; (3) *Present-Hedonistic*, a hedonistic, enjoyment- and pleasure-oriented attitude towards time without worrying about the future associated to low need for predictability, poor impulse control and increased novelty seeking; (4) *Present-Fatalistic*, a fatalistic, helpless, and hopeless attitude toward the future and life that is related to aggression, anxiety, and depression; and (5) *Future*, an orientation that includes the planning for and achievement of future goals and the tendency to postpone direct gratification in favor of long-term goals (*Zimbardo & Boyd, 1999*). Time perspective has also been found to influence cognitive well-being, that is, an individual's own evaluation of her/his life satisfaction (*Diener, 1984*). Whereas no single time perspective in itself fosters life satisfaction, it is predicted by a "balanced" time perspective: high values on past positive, present hedonistic and future perspectives, and low values on past negative and present fatalistic perspectives (*Boniwell et al., 2010*). Moreover, personal characteristics that allow the individual to adapt and flourish in life (i.e., psychological well-being; *Ryff, 1989*) seem to also be related to the same pattern of "balanced" time perspective (*Sailer et al., 2014*). Nevertheless, well-being
[1]Positive affect is a dimension that varies from pleasant engagement (e.g., enthusiastic and active) to unpleasant disengagement (e.g., sad and bored). The negative affect dimension, on the other hand, moves from unpleasant engagement (e.g., anger and fear) to pleasant disengagement (e.g., calm and serene) (*Watson & Clark, 1994*; *Watson, Clark & Tellegen, 1988*).

has been suggested to be a result of a complex interaction of a person's affective, cognitive, and social characteristics (*Cloninger, 2004*).

In accordance with such complex patterns between and within individuals, a correlational study found different patterns of time perspective to be associated to positive and negative affect.[1] Positive affect was positively related to the present hedonistic and future time perspective, but negatively related to the past negative and present fatalistic time dimensions (*Sailer et al., 2014*). Negative affect on the other hand was positively related to the past negative and present fatalistic time perspective dimensions and negatively related to the present hedonistic time perspective dimension (*Sailer et al., 2014*). Although these results give an indication on the association between individuals' time perspective and affectivity, the affective system is often described as a complex dynamic system composed of these two affectivity dimensions (i.e., positive affect and negative affect), which are independent of each other and regulate our approach and withdrawal behavior towards stimuli (e.g., *Watson, Clark & Tellegen, 1988*). Individuals characterized by high levels of positive affect exhibit a greater appreciation of life, more security, self-esteem and self-confidence (*Archer, Adolfsson & Karlsson, 2008*) they enjoy more social relations and assertiveness and are generally described as passionate, happy, energetic and alert (*Watson & Clark, 1984*; *Watson & Pennebaker, 1989*). In contrast, individuals characterized by high levels of negative affect experience greater stress and strain, anxiety and uncertainty over a wide range of circumstances and events over which they generally lack control (*Watson, Pennebaker & Folger, 1986*). The two affectivity dimensions are not only related to different behavior but are also influenced by the environment to different extent and have different genetic etiology (see *Cloninger & Garcia, 2015*). The independent inter-relationship of these two affectivity dimensions also implies that individuals do not only differ in affectivity between each other but also within themselves (*Garcia, 2011*; *Garcia et al., 2015*). If so, individuals might differ in the way they perceive time depending on their affective profile (i.e., different combinations of high/low positive/negative affect) and what's more, the way in which time perspective is related to well-being might be moderated by the individuals' own affective profile.

Previously, Archer and colleagues (e.g., *Archer, Adolfsson & Karlsson, 2008*; *Norlander, Bood & Archer, 2002*) conceptualized how individuals' differ, between and within, in levels of affectivity by incorporating different combinations of individuals' recalled experience of positive and negative affect, resulting in different "affective profiles": (i) high positive affect and low negative affect, characterizing a "self-fulfilling" profile, (ii) high positive affect and high negative affect, characterizing a "high affective" profile, (iii) low positive affect and low negative affect, characterizing a "low affective" profile, and (iv) low positive affect and high negative affect, characterizing a "self-destructive" profile. Individuals with high negative affect, particularly those with a self-destructive profile compared to individuals with a self-fulfilling profile, report lower well-being, higher psychological and somatic stress, low energy, lack of dispositional optimism, heightened pessimism, high levels of non-constructive perfectionism, depression and anxiety, lower levels of constructive coping and higher levels of maladaptive coping, total stress at the work-place, more Type A behavior, lack of emotional stability and partner relationships, and high levels of external locus of

control and impulsiveness (e.g., *Andersson-Arntén, 2009*, *Garcia, 2011*; *Garcia, MacDonald & Archer, 2015*; *Schütz, 2015*; *Norlander, Bood & Archer, 2002*; *Norlander, Johansson & Bood, 2005*; *Bood, Archer & Norlander, 2004*). Thus, individuals with a self-destructive profile can be expected to have a less "balanced" time perspective compared to individuals with any of the other profiles.

The affective profiles model allows the comparison between individuals who differ in their level of experienced affect in both dimensions, but also the comparison of individuals who match each other in one dimension and differ in their experience in the other affect dimension (i.e., allowing a within-individual comparison). For example, when individuals with a low affective profile are compared to their diametric opposites (i.e., individuals with a high affective profile), they show higher levels of somatic stress when doing a stressful task (*Norlander, Johansson & Bood, 2005*). This may be because, in contrast to individuals with a low affective profile, individuals with a high affective profile experience high positive affect, which may neutralize their experience of high negative affect and therefore, reduce stress (*Fredrickson, 2006*; *Garcia & Siddiqui, 2009a*). Nevertheless, individuals with any of these two profiles do not differ in life satisfaction between each other (*Garcia & Siddiqui, 2009a*; *Garcia & Siddiqui, 2009b*). This suggests that for individuals with a low affective profile, low levels of stress and high levels of life satisfaction are linked to their experience of low negative affect, while for individuals with a high affective profile this very same experience (i.e., low stress and high life satisfaction) is linked to high levels of positive affect (cf. *Garcia, 2011*; *Schütz, 2015*).

In addition, when individuals with a low affective profile are compared to individuals to whom they only partially differ in affectivity levels (i.e., self-destructive and self-fulfilling), individuals with a low affective profile report higher life satisfaction than individuals with a self-destructive profile and equally high levels of life satisfaction and equally low levels of stress as individuals with a self-fulfilling profile (*Garcia, 2011*). In other words, although both of these profiles (i.e., low affective and self-destructive) are characterized by low levels of positive affect, individuals with a low affective profile (low positive affect/low negative affect) are more satisfied with their life than individuals with a self-destructive profile (low positive affect/high negative affect). Individuals with a low affective profile (low positive affect/low negative affect) are also as satisfied with their life as individuals with a self-fulfilling profile (high positive affect/low negative affect), although the latter experience more positive affect.

In sum, depending on their profile, individuals are able to regulate their well-being, probably by specific strategies that fit their profile to maintain homeostasis in their affective system (cf. *Garcia et al., 2010*). If so, different time perspectives might influence individuals' life satisfaction and psychological well-being depending on their affective profile. The present study investigated differences in time perspective and well-being between individuals with distinct affective profiles. We expected individuals with a self-fulfilling profile to be more "balanced" in their time perspective: more positive and less negative about their past, more hedonistic and less fatalistic about their present, and more future oriented. We also addressed the question whether or not the effect of the time

perspective dimensions on psychological well-being and life satisfaction is moderated by the individual's type of profile.

## METHOD

### Ethical statement

After consulting with the Network for Empowerment and Well-Being's Review Board we arrived at the conclusion that the design of the present study (e.g., all participants' data were anonymous and will not be used for commercial or other non-scientific purposes) required only informed consent from the participants.

### Participants and procedure

The present study was based on a sample of 720 participants with an age mean of 25.25 ± 11.73 (males = 247, females = 473, and seven participants who didn't report their gender). They were students at one University and pupils at two high schools in the West of Sweden. All participants were informed that their participation was voluntary and anonymous. They were presented with a battery of instruments used to collect the relevant measures in the following order: background, time perspective, temporal satisfaction with life, psychological well-being, and affect.

### Measures

#### Affect

The Positive Affect and Negative Affect Schedule (*Watson, Clark & Tellegen, 1988*) assesses the affective component of subjective well-being by requiring participants to rate to what extent (1 = *very slightly*, 5 = *extremely*) during the last few weeks they have experienced 10 positive and 10 negative affective states. The positive affect scale includes adjectives such as strong, proud, and interested; and the negative affect scale includes adjectives such as afraid, ashamed, and nervous. The Swedish version has been used in previous studies with good psychometric properties (*Cronbach's α* between .88 and .90; e.g., *Garcia, Nima & Kjell, 2014*; *Nima, Archer & Garcia, 2012*; *Nima, Archer & Garcia, 2013*; *Nima et al., 2013*; *Schütz, Archer & Garcia, 2013*). *Cronbach's α* in the present study was .86 for positive affect and .85 for negative affect.

#### Time perspective

The Zimbardo Time Perspective Inventory (*Zimbardo & Boyd, 1999*) consists of 56 items that measure the following five time dimensions: Past Positive (e.g., "It gives me pleasure to think about my past"), Past Negative (e.g., "I think about the good things that I have missed out on in my life"), Present Hedonistic (e.g., "Taking risks keeps my life from becoming boring"), Present Fatalistic (e.g., "Fate determines much in my life"), and Future (e.g., "I believe that a person's day should be planned ahead each morning"). The Swedish version has been used in previous studies and showed good psychometric properties (*Cronbach's α* between .68 and .87; e.g., *Sailer et al., 2014*) and its psychometric properties have been validated in many different languages (*Milfont et al., 2008*; *Liniauskaite & Kairys, 2009*; *Díaz-Morales, 2006*). Cronbach's α in the present study was .72 for Past Positive, .85 for Past Negative, .76 for Present Hedonistic, .63 for Present Fatalistic, and .70 for Future.
### Temporal life satisfaction

The Temporal Satisfaction With Life Scale (*Pavot, Diener & Suh, 1998*) comprises 15-items rated on a 7-point Likert scale (1 = *strongly disagree*, 7 = *strongly agree*) assessing past (e.g., "If I had my past to live over, I would change nothing"), present (e.g., "I would change nothing about my current life"), and future life satisfaction (e.g., "There will be nothing that I will want to change about my future"). The Swedish version of the instrument has been used in previous studies (*Cronbach's α* between .88 and .93; *Sailer et al., 2014*; *Garcia, Rosenberg & Siddiqui, 2011*). Cronbach's α in the present study was .92 for the whole scale.

### Psychological well-being

The Psychological Well-Being scale, short version (*Clarke et al., 2001*) comprises 18 items including three items for each of the six dimensions. These dimensions are: self-acceptance (e.g., "I like most aspects of my personality"), personal growth (e.g., "For me, life has been a continuous process of learning, changing, and growth"), purpose in life ("Some people wander aimlessly through life, but I am not one of them"), environmental mastery (e.g., "I am quite good at managing the responsibilities of my daily life"), autonomy (e.g., "I have confidence in my own opinions, even if they are contrary to the general consensus"), and positive relations with others (e.g., "People would describe me as a giving person, willing to share my time with others"). The Swedish version has been used in previous studies (e.g., *Garcia, 2011*; *Garcia, 2014*). Since the subscales have been found to have low reliability, the total psychological well-being score (i.e., the sum of the 18 items) is recommended as a better and more reliable measure (*Garcia & Siddiqui, 2009b*). A Cronbach's α of .78 was obtained for the total psychological well-being score in the present study.

## Statistical treatment

The affective profiles were created by dividing self-reported positive affect and negative affect scores into high and low using a median split (*Norlander, Bood & Archer, 2002*). This resulted in the following affective profiles: 222 "self-destructive" (low positive and high negative affect), 131 "low affective" (low positive and low negative affect), 150 "high affective" (high positive and high negative affect) and 217 "self-fulfilling" (high positive and low negative affect).

### Missing data

The majority of missing data was found to be missing completely at random using *Little's Chi-Square* test; ($\chi 2 = 30.10$ ($df = 28$, $p = .36$) for self-destructive men, $\chi 2 = 17.54$ ($df = 9$, $p = .04$) for low affective men, $\chi 2 = 26.79$ ($df = 20$, $p = .14$) for high affective men, $\chi 2 = 17.61$ ($df = 15$, $p = .28$) for self-fulfilling men, $\chi 2 = 37.87$ ($df = 37$, $p = .43$) for self-destructive women, $\chi 2 = 22.69$ ($df = 26$, $p = .65$) for low affective women, $\chi 2 = 26.14$ ($df = 27$, $p = .51$) for high affective women and $\chi 2 = 57.24$ ($df = 28$, $p = .001$) for self-fulfilling women. The Expectation-Maximization Algorithm[2] was used to replace missing values.

### Normality of sampling distributions of means

Our sample size of 720 participants included over 20 cases for each cell. Therefore, we anticipated normality of sampling distributions of means. Indeed, according to the Central

[2]The Expectation-Maximization Algorithm is one of the most common algorithms to estimate the parameters (e.g., means and standard deviations) of a statistical model given data (https://en.wikipedia.org/wiki/Maximum_likelihood). In other words, this specific algorithm is an iterative method for finding maximum likelihood or maximum a posteriori estimates of parameters in statistical models, where the model depends on unobserved latent variables. Missing data analyses using Monte Carlo technique show that the Expectation-Maximization Algorithm is more reliable for missing-data imputation when compared to pairwise and listwise deletion (*Malhotra, 1987*; *Graham & Donaldson, 1993*). Hence, we found it as an appropriate method in the present study.

Limit Theorem, with sufficiently large sample sizes, sampling distributions of means are normally distributed regardless of the distributions of variables. (see *Tabachnick & Fidell, 2007*, p. 78). In other words, our data met the assumptions necessary to conduct a MANOVA.

### Univariate outliers and normality

In order to determine and reduce the impact of variables with univariate outliers within the affective profiles we first standardized the scores by subtracting the mean from the individual's score and then dividing by the standard deviation. We then checked if any cases had larger standardized scores than $\pm3.29$, as recommended by *Tabachnick & Fidell (2007)*. One outlier was detected in temporal satisfaction with life, one in past negative, one in present fatalistic and two in future (i.e., standardized scores larger than $\pm3.29$). These outlier scores were changed to the next highest non-outlier score $+1$, as described by *Tabachnick & Fidell* (*2007*, p. 77).

All following analyses were computed with these replaced values for the outliers and the original raw-scores. The dependent variables (the five time perspective dimensions, psychological well-being and temporal satisfaction) per affective profile were normally distributed with a skewness between .07 and $-.77$ and a kurtosis between .01 and $-.84$. Because our sample size is relatively large, these values are reasonable (see *Tabachnick & Fidell, 2007*, p. 80). Visual inspection indicated no threats to linearity or homoscedasticity on the dependent variables (i.e., time perspective and well-being) for each affective profile. Thus, the assumptions were met to conduct the SEM.

### Multivariate outliers

The five time perspective dimensions, psychological well-being and temporal satisfaction with life were checked for multivariate outliers within the affective profiles. The multivariate outlier detection by Mahalanobis distance identified three multivariate outliers, which were replaced as described above (*Tabachnick & Fidell, 2007*, p. 76).

### Multicollinearity and singularity

The correlations between dependent variables were all below $-.59$. These correlations, for each profile, were below $-.54$. Therefore, multicollinearity or singularity was judged as unlikely to be present or a problem (see *Tabachnick & Fidell, 2007*, p. 88, who recommend .90 as threshold).

Differences in psychological well-being and temporal satisfaction with life between affective profiles were investigated using a MANOVA. Psychological well-being and temporal satisfaction with life served as dependent variables, affective profiles were the independent variables. A second MANOVA investigated differences between affective profiles in the five dimensions of time perspective. Here, the mean scores on each of the time perspective dimension scale served as dependent variables and affective profile as independent variable. Each MANOVA, if significant regarding *Pillai's* criterion, was followed up by ANOVA to test differences between individuals with distinct profiles on each of the dependent variables and then we conducted post-hoc tests with Bonferroni correction to investigate which profiles differed from each other.

### Homogeneity of variance–covariance matrices

The *Box's M* test was significant at $p < .001$ for the first MANOVA (i.e., the analysis investigating differences in psychological well-being and temporal satisfaction with life between affective profiles) and at $p < .02$ (see *Huberty & Petoskey, 2000*, who suggest that a $p$ value higher than the cut-off of $p = .005$ does not violate the assumption of homogeneity of variance–covariance matrices) for the second MANOVA (i.e., the analysis investigating differences between affective profiles in the 5 dimensions of time perspective). Nevertheless, the groups in each profile are relatively large and there are only small group size differences (with a ratio of 1.69:1 regarding profiles the largest group was 222 self-destructive profile and the smallest was 131 low affective profile). As a preliminary check for robustness, large groups have larger variances and covariances in the dependent variables, compared to small groups with smaller sizes; however, in our data there were only small differences in the sizes of the variances and covariances. For example, regarding variances for the first MANOVA the ratio of largest (.27) to smallest (.10) variance was 2.70:1 (temporal satisfaction with life). Regarding variances for the second MANOVA the ratio of largest (.31) to smallest (.22) variance was 1.41:1 (present fatalistic). MANOVA makes the assumption that the within-group covariance matrices are equal. If the design is balanced so that there are an equal number of observations in each cell, the robustness of the MANOVA tests is guaranteed. Thus, the assumptions of homogeneity of variance and covariance matrices were met for the conduction of MANOVAs (see *Tabachnick & Fidell, 2007*). Moreover, we used Pillai's criterion instead of Wilks' lambda because Pillai's criterion is more robust, appropriate, and more stringent criterion against heterogeneity of variance–covariance (see *Tabachnick & Fidell, 2007*, p. 252).

### Residuals of the covariances among observed variables in the SEM

All the residual covariances and standardized residual covariances among observed variables for each profile were zero, with the exception of covariances between psychological well-being and temporal life satisfaction which were between .10 for residual covariance and 3.28 for standardized residual covariance for each affective profile. Nevertheless, the residuals for both variables were still centered around zero and the sample size used here is relatively large, thus, our multi-group moderation model fits the data reasonably well and the residuals were considered symmetrical (see *Tabachnick & Fidell, 2007*, p. 684).

## RESULTS

### Differences in psychological well-being and temporal satisfaction with life between affective profiles

The affective profiles had a significant effect on the psychological well-being and temporal satisfaction with life ($F(6, 1,432) = 43.80$, $p < .001$, *Pillai's Trace* $= .31$, Observed Power $= 1.00$). The groups differed in psychological well-being ($F(3, 716) = 59.57$, $p < .001$, Observed Power $= 1.00$) and in temporal satisfaction with life ($F(3, 716) = 77.37$, $p < .001$, Observed Power $= 1.00$).

Individuals with a self-fulfilling profile scored higher in psychological well-being and in temporal satisfaction with life than individuals with any of the other profiles. Individuals

**Table 1 Mean scores and standard deviation (*sd*) in psychological well-being, temporal life satisfaction and the time perspective dimensions for each affective profile.**

| | Self-destructive $n = 222$ | Low-affective $n = 131$ | High-affective $n = 150$ | Self-fulfilling $n = 217$ |
|---|---|---|---|---|
| Psychological well-being | $3.86 \pm .49$ | $4.18 \pm .58^{a***}$ | $4.31 \pm .54^{a***}$ | $4.57 \pm .63^{a,b,c***}$ |
| Temporal life satisfaction | $3.42 \pm 1.11$ | $4.36 \pm 1.03^{a***}$ | $4.38 \pm 1.07^{a***}$ | $4.89 \pm .89^{a,b,c***}$ |
| Past negative | $3.29 \pm .66^{b,c,d***}$ | $2.57 \pm .62$ | $2.94 \pm .69^{b,d***}$ | $2.44 \pm .64$ |
| Past positive | $3.20 \pm .64$ | $3.34 \pm .60$ | $3.45 \pm .63^{a**}$ | $3.49 \pm .63^{a***}$ |
| Present fatalistic | $2.58 \pm .52^{b,d***,c**}$ | $2.32 \pm .47$ | $2.38 \pm .56^{d**}$ | $2.19 \pm .46$ |
| Present hedonistic | $3.09 \pm .45$ | $3.00 \pm .46$ | $3.22 \pm .45^{b**}$ | $3.17 \pm .53^{b*}$ |
| Future | $3.14 \pm .48$ | $3.14 \pm .45$ | $3.39 \pm .45^{a,b***}$ | $3.38 \pm .47^{a,b***}$ |

**Notes.**

Values represent mean scores $\pm$ *sd*.

[*]$p < .05$.

[**]$p < .01$.

[***]$p < .001$.

[a]Higher compared to the self-destructive.

[b]Higher compared to the low affective.

[c]Higher compared to the high affective.

[d]Higher compared to the self fulfilling.

with a high affective and low affective profile scored higher in both temporal satisfaction with life and psychological well-being compared to individuals with a self-destructive profile (see details in Table 1).

## Differences in the five dimensions of time perspective between affective profiles

The affective profiles had a significant effect on the time perspective dimensions ($F(15, 2, 142) = 18.18$, $p < .001$, *Pillai's Trace* $= .35$, Observed Power $=1.00$). The groups differed in the past negative ($F(3, 716) = 69.84$, $p < .001$, Observed Power $= 1.00$), past positive ($F(3, 716) = 9.40$, $p < .001$, Observed Power $= 1.00$), present fatalistic ($F(3, 716) = 22.30$, $p < .001$, Observed Power $= 1.00$), present hedonistic ($F(3, 716) = 5.76$, $p < .001$, Observed Power $= .95$), and future ($F(3, 716) = 16.69$, $p < .001$, Observed Power $= 1.00$) dimensions. Compared to individuals with any of the other profiles, individuals with a self-destructive profile scored higher in past negative and present fatalistic time perspective. Individuals with a self-fulfilling profile and a high affective profile scored higher in past positive time perspective compared to individuals with a self-destructive profile. Individuals with a self-fulfilling profile and a high affective profile scored higher in present hedonistic as compared to individuals with a low affective time perspective. Individuals with a self-fulfilling profile and a high affective profile scored higher in the future dimension compared to individuals with a self-destructive profile and a low affective profile. See Table 1 for the details in which the results from the post hoc tests, Bonferroni correction: $p = .05 \div 5 = .01$, are presented.

## Multi-group moderation analysis

To investigate which of the time perspective dimensions were related to both psychological well-being and temporal satisfaction with life we performed a path analysis, using AMOS

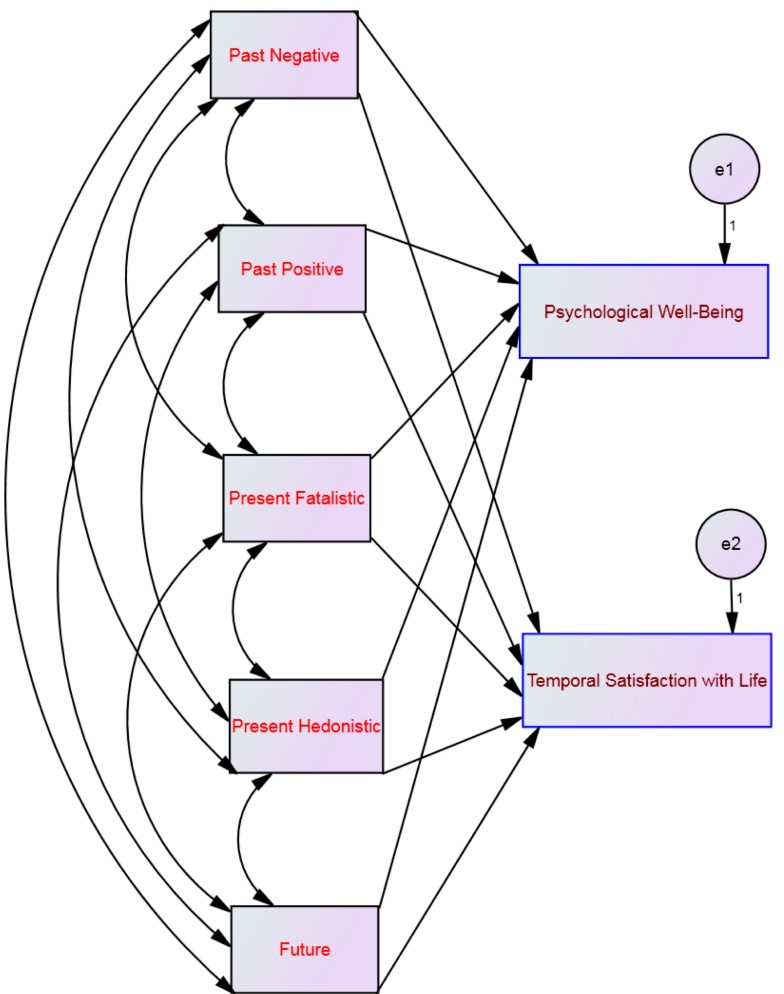

**Figure 1** Hypothesized structural equation model using the time perspective dimensions as predictors of both psychological well-being and temporal satisfaction with life.

(version 20)—in order to estimate interaction/moderation effects between affective profiles as the moderator, time perspective dimensions as independent variables, and psychological well-being and temporal satisfaction with life as the outcome (see Fig. 1). The structural equation model of multi-group analysis showed a *Chi-square*= 23.22; *DF* = 4; *p* < .001. The large sample in our present study ($N = 720$) may influence the *Chi-square* value to be significant (see *Tabachnick & Fidell, 2007*, p. 695). However, the path model yielded a good fit, as indicated by comparative fit index = .98; goodness of fit index = .99; incremental fit index = .98, normed fit index = .97 and root mean square error of approximation = .08.

Four multi-group moderation analyses, one for each profile, showed that 16%–33% of the variance of psychological well-being and 29%–40% of the variance of temporal satisfaction with life could be explained by the five time perspective dimensions (see Table 2). Specifically, psychological well-being was significantly predicted by past positive and present hedonistic across all affective profiles (see Figs. 2–5). This suggests that the type of affective profile does not moderate the influence of these two time perspective

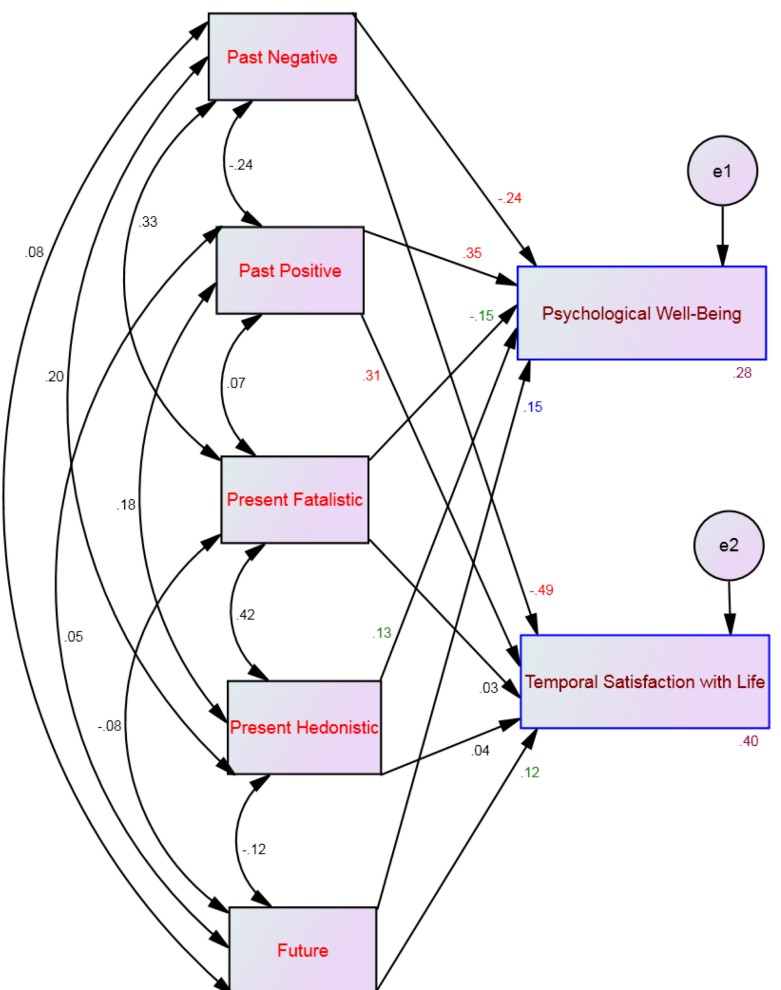

**Figure 2** **SEM for the self-destructive profile showing all correlations (between time perspective dimensions) and all paths (from time perspective to well-being) and their standardized parameter estimates.** Note: *Chi-square = 23.22; df = 4; p < .001; comparative fit index = .98; goodness of fit index = .99; incremental fit index = .98, normed fit index = .97* and *root mean square error of approximation = .08.* Red standardized parameter estimates of regression weights are significant at the *p < .001* level, blue standardized parameter estimates of regression weights are significant at the *p < .01* level and green standardized parameter estimates of regression weights are significant at the *p < .05* (*n = 222*).

dimensions on psychological well-being. For individuals with a self-destructive profile, psychological well-being was significantly predicted by past negative, present fatalistic, and future (see Fig. 2). Among individuals who experience high levels of positive affect (i.e., high affective and self-fulfilling), psychological well-being was significantly predicted by the present fatalistic dimension (see Figs. 4 and 5). Temporal satisfaction with life was significantly predicted by past negative and past positive across all affective profiles (see Figs. 2–5). This suggests that the type of affective profile does not moderate the influence of these two time perspective dimensions on temporal satisfaction with life. Nevertheless, for individuals with a self-destructive profile, temporal satisfaction with life was predicted by the future time perspective dimension (see Fig. 2).

**Table 2** Structural coefficients for the multi-group moderation analyses using the type of affective profile as the moderator, the time perspective dimensions as predictors and both psychological well-being and temporal satisfaction with life as the outcomes.

| Predictor | Outcome | β | SE | B | P |
|---|---|---|---|---|---|
| *Self-destructive = 222* | | | | | |
| **Past negative** | | **−.18** | **.05** | **−.24** | **<.001** |
| **Past positive** | | **.27** | **.05** | **.35** | **<.001** |
| **Present fatalistic** | Psychological well-being | **−.14** | **.06** | **−.15** | **<.005** |
| **Present hedonistic** | | **.15** | **.07** | **.13** | **<.05** |
| **Future** | | **.16** | **.06** | **.15** | **<.01** |
| $R^2$ | .28 | | | | |
| **Past negative** | | **−.82** | **.10** | **−.49** | **<.001** |
| **Past positive** | | **.53** | **.10** | **.31** | **<.001** |
| Present fatalistic | Temporal satisfaction | .06 | .13 | .03 | .63 |
| Present hedonistic | | .10 | .14 | .04 | .49 |
| **Future** | | **.28** | **.12** | **.12** | **<.05** |
| $R^2$ | .40 | | | | |
| *Low-affective n = 131* | | | | | |
| Past negative | | −.06 | .08 | −.06 | .45 |
| **Past positive** | | **.42** | **.08** | **.44** | **<.001** |
| Present fatalistic | Psychological well-being | −.03 | .10 | −.03 | .75 |
| **Present hedonistic** | | **.29** | **.10** | **.23** | **<.01** |
| Future | | −.09 | .10 | −.07 | .35 |
| $R^2$ | .28 | | | | |
| **Past negative** | | **−.69** | **.14** | **−.41** | **<.001** |
| **Past positive** | | **.61** | **.14** | **.35** | **<.001** |
| Present fatalistic | Temporal satisfaction | .12 | .18 | .05 | .52 |
| Present hedonistic | | .10 | .18 | .04 | .57 |
| Future | | −.21 | .18 | −.09 | .23 |
| $R^2$ | .30 | | | | |
| *High-affective n = 150* | | | | | |
| Past negative | | −.02 | .06 | −.03 | .74 |
| **Past positive** | | **.35** | **.06** | **.41** | **<.001** |
| **Present fatalistic** | Psychological well-being | **−.30** | **.07** | **−.32** | **<.001** |
| **Present hedonistic** | | **.37** | **.09** | **.31** | **<.001** |
| Future | | .13 | .08 | .11 | .10 |
| $R^2$ | .33 | | | | |
| **Past negative** | | **−.64** | **.13** | **−.41** | **<.001** |
| **Past positive** | | **.39** | **.13** | **.23** | **<.001** |
| Present fatalistic | Temporal satisfaction | .04 | .15 | .02 | .78 |
| Present hedonistic | | .19 | .19 | .08 | .33 |
| Future | | .30 | .17 | .13 | .07 |
| $R^2$ | .29 | | | | |
| *Self-fulfilling n = 217* | | | | | |
| Past negative | | −.02 | .07 | −.02 | .47 |

**Table 2** (*continued*)

| Predictor | Outcome | β | SE | B | P |
|---|---|---|---|---|---|
| **Past positive** | | **.30** | **.07** | **.30** | **<.001** |
| **Present fatalistic** | Psychological well-being | **−.20** | **.10** | **−.14** | **<.05** |
| **Present hedonistic** | | **.22** | **.08** | **.19** | **<.01** |
| Future | | .11 | .09 | .09 | .19 |
| **$R^2$** | .16 | | | | |
| **Past negative** | | **−.68** | **.08** | **−.48** | **<.001** |
| **Past positive** | | **.39** | **.08** | **.28** | **<.001** |
| Present fatalistic | Temporal satisfaction | .08 | .12 | .04 | .49 |
| Present hedonistic | | .12 | .10 | .07 | .25 |
| Future | | .09 | .10 | .05 | .42 |
| **$R^2$** | .36 | | | | |

**Notes.**
 Significant regression weights are shown in bold type.

## DISCUSSION

This study revealed differences in time perspective and well-being depending on an individual's affective profile (for a summary of the results see Fig. 6). By looking at the differences between individuals at the diametrical ends of the model we first found that individuals with a self-fulfilling profile (i.e., high positive and low negative affect), compared to individuals with a self-destructive profile (i.e., low positive and high negative affect), scored high on psychological well-being, high on temporal life satisfaction, high on the past positive and future time perspective dimensions, and low in both the past negative and present fatalistic time perspective dimensions (see Fig. 6, horizontal black arrows). This fits the description of a "balanced" time perspective that promotes high levels of well-being (*Boniwell et al., 2010*). Also being diametrically different to individuals with a low affective profile (low positive and low negative affect), individuals with a high affective profile (high positive and high negative affect) scored higher on the past negative, the present hedonistic and the future time perspective dimensions (see Fig. 6, vertical black arrows). As in earlier studies (e.g., *Garcia & Siddiqui, 2009a*), no differences in well-being were found between individuals with high and low affective profiles. Nevertheless, individuals with any of these two profiles scored higher on both psychological well-being and temporal life satisfaction when compared to those with a self-destructive profile. Hence, a low level of positive affect together with a high level of negative affect appears to be detrimental for psychological well-being and life satisfaction.

One of the strengths of the affective profiles model is that it allows the comparison of people who differ in one affectivity dimension while keeping the other constant. In this way we get to observe associations within this complex adaptive system (cf. *Cloninger & Garcia, 2015*). For example, decreases in a negative view of the past (i.e., the past negative time perspective dimension) might lead to high positive affect when negative affect is high (see grey arrows in Fig. 6: self-destructive vs. high affective) but to low negative affect when positive affect is either high (see grey arrows in Fig. 6: high affective vs. self-fulfilling) or

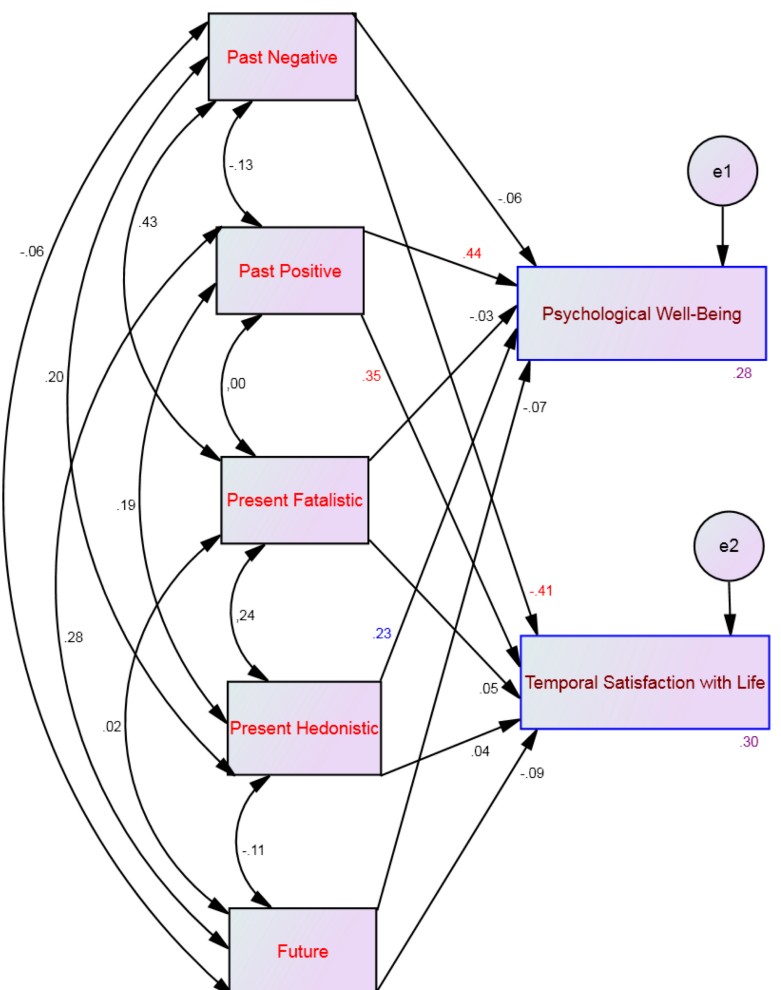

**Figure 3** **SEM for the low affective profile showing all correlations (between time perspective dimensions) and all paths (from time perspective to well-being) and their standardized parameter estimates.** Note: *Chi-square* = 23.22; *df* = 4; *p* < .001; *comparative fit index* = .98; *goodness of fit index* = .99; *incremental fit index* = .98, *normed fit index* = .97 and *root mean square error of approximation* = .08. Red standardized parameter estimates of regression weights are significant at the *p* < .001 level, blue standardized parameter estimates of regression weights are significant at the *p* < .01 level and green standardized parameter estimates of regression weights are significant at the *p* < .05 (*n* = 131).

low (see grey arrows in Fig. 6: self-destructive vs. low affective). Increases in the future perspective dimension seem to only be associated with increases in positive affect; both when negative affect is high (see grey arrows in Fig. 6: self-destructive vs. high affective) and when negative affect is low (see grey arrows in Fig. 6: low affective vs. self-fulfilling). In contrast, increases in the present hedonistic dimension seem to lead to higher levels of positive affect only when negative affect is low (see grey arrows in Fig. 6: low affective vs. self-fulfilling). Low levels in negative affect in turn were associated to decreases in the past negative time perspective dimension. In other words, to live happy in the present we need to let go of our past. The act of letting go of struggles is indeed one of the first steps of self-aware knowledge that is part of the Science of Well-Being (see *Cloninger,*

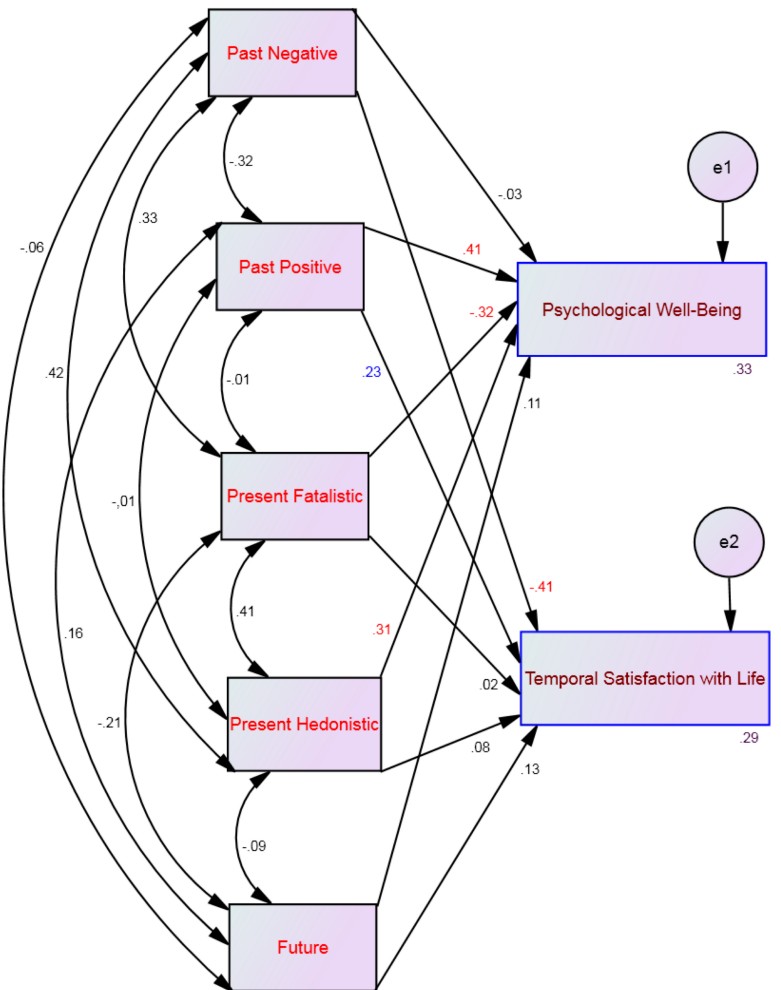

**Figure 4** **SEM for the high affective profile showing all correlations (between time perspective dimensions) and all paths (from time perspective to well-being) and their standardized parameter estimates.**
Note: *Chi-square* = 23.22; *df* = 4; *p* < .001; *comparative fit index* = .98; *goodness of fit index* = .99; *incremental fit index* = .98, *normed fit index* = .97 and *root mean square error of approximation* = .08. Red standardized parameter estimates of regression weights are significant at the *p* < .001 level, blue standardized parameter estimates of regression weights are significant at the *p* < .01 level and green standardized parameter estimates of regression weights are significant at the *p* < .05 (*n* = 150).

*2004*). All these complex interactions give a picture of how time perspective dimensions are associated to the affectivity system. Next we discuss how these dimensions predict well-being depending on the person's own affective profile.

Interestingly, certain time perspective dimensions influenced well-being depending on the person's type of affective profile. Indeed, moderation analysis showed that the past positive and the present hedonistic time perspectives were positively associated to psychological well-being among individuals with any type of affective profile, while the present fatalistic dimension was negatively associated to psychological well-being in three out of the four affective profile groups—the exception was for individuals with a low affective profile. Individuals with a low affective profile have been found to downplay

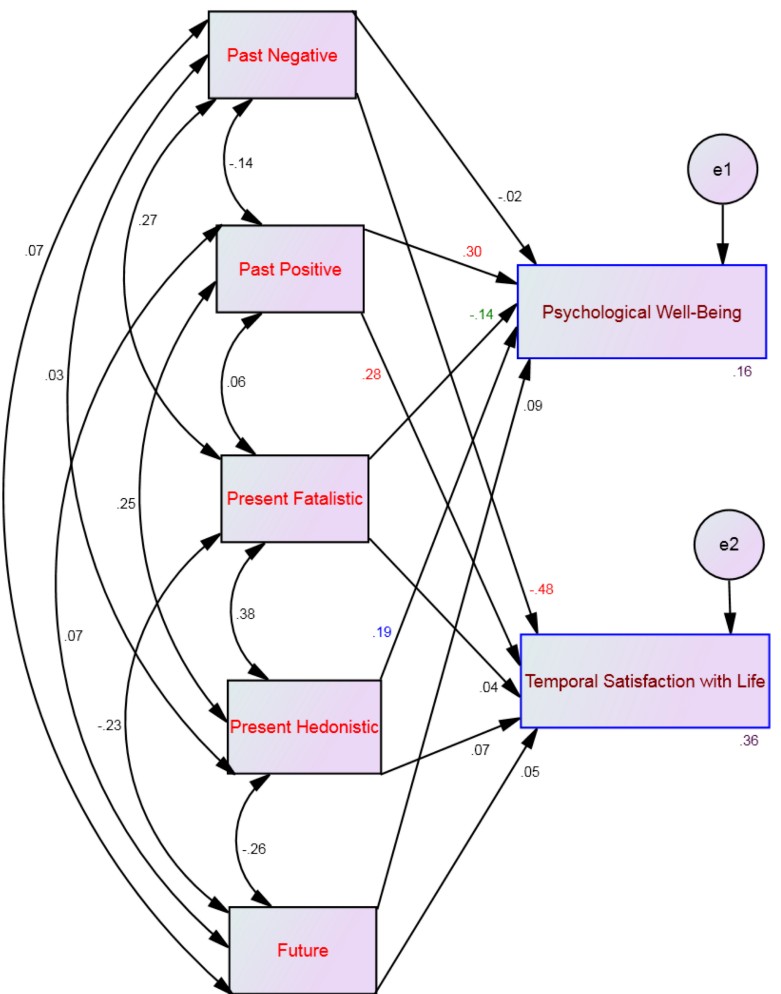

**Figure 5** SEM for the self-fulfilling profile showing all correlations (between time perspective dimensions) and all paths (from time perspective to well-being) and their standardized parameter estimates.
Note: *Chi-square* = 23.22; *df* = 4; *p* < .001; *comparative fit index* = .98; *goodness of fit index* = .99; *incremental fit index* = .98, *normed fit index* = .97 and *root mean square error of approximation* = .08. Red standardized parameter estimates of regression weights are significant at the *p* < .001 level, blue standardized parameter estimates of regression weights are significant at the *p* < .01 level and green standardized parameter estimates of regression weights are significant at the *p* < .05 (*n* = 217).

their emotions by either neutralizing positive and negative stimuli or, when faced with many positive things in life, to value neutral stimuli as more negative (*Garcia & Siddiqui, 2009a*; *Garcia & Siddiqui, 2009b*; *Garcia et al., 2010*). Individuals with a low affective profile probably use these strategies in order to stay in an affective state that is more in tune to their profile. Together with our findings here (i.e., the negative effect of the present fatalistic time perspective on well-being among people with any profile but among individuals with a low affective profile), this earlier findings might suggest that individuals with a low affective profile achieve homeostasis through being fatalistic of their present, that is, seeing their life path as controlled by external forces, avoiding to worry about the future because they also see it as uncontrollable, believing in luck or fate rather than

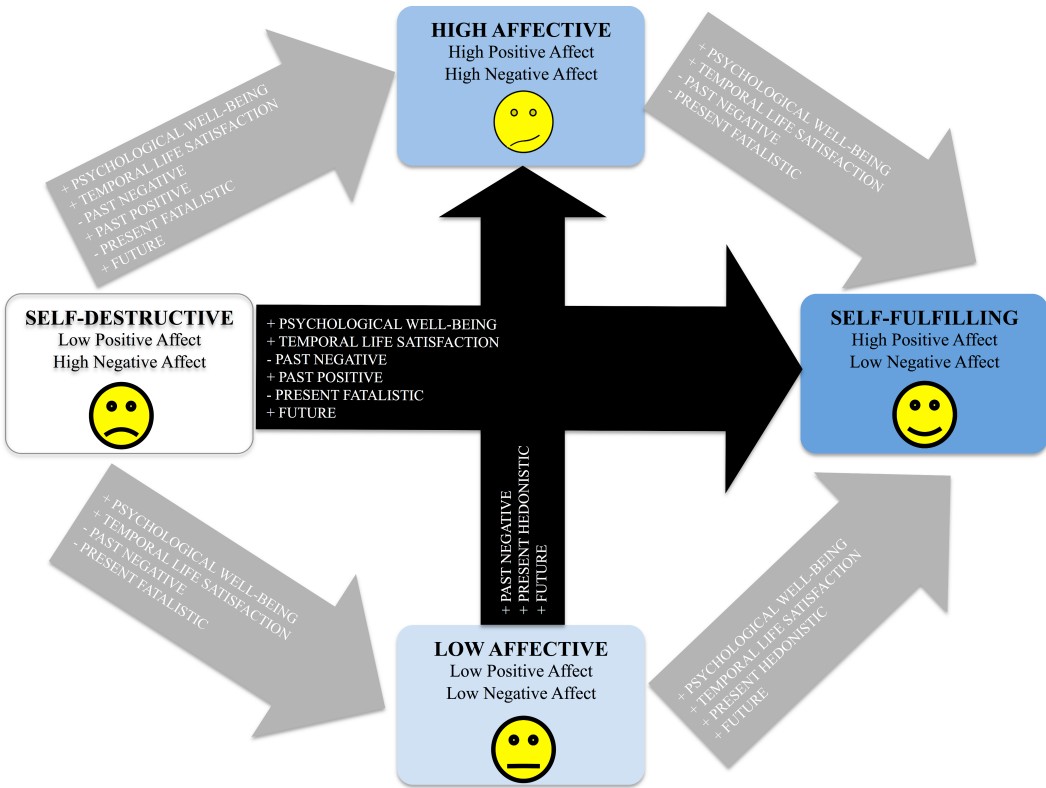

**Figure 6** **Differences between (black arrows) and within (grey arrows) individuals.** Differences (black arrows) found between individuals with affective profiles that are at their extremes of the model: self-destructive versus self-fulfilling (low–high positive affect, high–low negative affect) and low affective versus high affective (low–high positive affect, low–high negative affect). Differences (grey arrows) found when individuals were matched in one affective dimension, and differed in the other (i.e., within differences): self-destructive versus high affective (matching: high–high negative affect, differing: low–high positive affect), self-destructive versus low affective (matching: low–low positive affect, differing: high–low negative affect), high affective versus self-fulfilling (matching: high–high positive affect, differing: high–low negative affect), and low affective versus self-fulfilling (matching: low–low negative affect, differing: low–high positive affect). Note. Reprinted with permission from Well-Being and Human Performance Sweden AB.

hard work, and avoid setting goals. This strategy does indeed help individuals with a low affective profile to prevent unhappiness (i.e., low levels of negative affect) and is certainly in line with how their affectivity system dynamically regulates itself (cf. self-regulatory theory; *Higgins, 2001*). In other words, by being fatalistic about their present they prevent becoming disappointed and just the absence of that possible disappointment makes them feel satisfied with their life (*Garcia et al., 2010*; see also *Fredrickson, 2006*; *Garcia & Siddiqui, 2009a*; *Ramsey et al., 2016*). Of course, at the same time the usage of this strategy limits their experience of positive emotions, which might explain why they are not as satisfied with life as individuals with a self-fulfilling profile. The pattern that emerges for temporal life satisfaction differs from the one for psychological well-being. Temporal life satisfaction was associated positively with past positive and negatively with past negative for all four profiles. Suggesting that these specific time perspective dimensions are equally important

for life satisfaction, independently of the individual's type of affective profile. In other words, the associations between nostalgic and positive attitudes towards the past and/or a general negative view of the past are both associated to life satisfaction regardless of how the individual's affectivity system is structured (i.e., high/low positive/negative affect). Indeed, others have suggested that the ability to let go of past struggles is related to feelings of hope that one can manage the present and the future (i.e., self-directedness; *Cloninger, 2004*). Feelings of hope, although positive, are not embedded in our affectivity system; they are rather associated to frontal lobe activity (*Cloninger, 2004*). In addition, for individuals with a self-destructive profile the future time perspective dimension was also associated to high levels of life satisfaction. Thus, suggesting a unique association among individuals with a self-destructive profile' life satisfaction and their ability to plan and achieve future goals and their tendency to postpone direct gratification in favor of long-term goals.

## Limitations

Time perspective manipulation has been reported to influence experienced affect (*Murgraff et al., 1999*; *Strack, Schwarz & Gschneidinger, 1985*) and vice versa. The present analysis of time perspective, however, presents an affective profile background derived from healthy volunteers. Those individuals presenting less healthy profiles, such as the self-destructive profiles, may arise from a "prodromal" phase of affective ill-being or a "past-experienced" affective condition. Without repeated measures or a sub-longitudinal analysis, the status of differential time perspectives over the affective profiles remains uncertain. Moreover, one important limitation is the fact that the sample was constituted of students and pupils.

## Concluding remarks

The interactions found here go beyond the postulation suggesting that the only way of promoting well-being, at least with regard to time dimensions, is a "balanced" time perspective. Instead, we present a more person-centered approach to achieve higher levels of emotional, cognitive, and psychological well-being. We suggest that future research might consider personality profiles to address the question of how interventions might affect a person's outlook on life (cf. *Garcia & Rosenberg, 2016*; *Cloninger & Zohar, 2011*).

### Funding

The authors received no funding for this work.

### Competing Interests

Danilo Garcia is the Director of the Blekinge Center of Competence, which is the Blekinge County Council's research and development unit. Ali Al Nima is a researcher and statistician at the Center. The Center works on innovations in public health and practice through interdisciplinary scientific research, person-centered methods, community projects, and the dissemination of knowledge in order to increase the quality of life of the habitants of the county of Blekinge, Sweden.

## Author Contributions

- Danilo Garcia conceived and designed the experiments, performed the experiments, analyzed the data, wrote the paper, prepared figures and/or tables, reviewed drafts of the paper.
- Uta Sailer and Trevor Archer wrote the paper, reviewed drafts of the paper.
- Ali Al Nima analyzed the data, wrote the paper, prepared figures and/or tables, reviewed drafts of the paper.

## Human Ethics

The following information was supplied relating to ethical approvals (i.e., approving body and any reference numbers):

After consulting with the Network for Empowerment and Well-Being's Review Board we arrived at the conclusion that the design of the present study (e.g., all participants' data were anonymous and will not be used for commercial or other non-scientific purposes) required only informed consent from the participants.

## Data Availability

The raw data is available upon request to the Network for Empowerment and Well-Being, lead researcher Danilo Garcia: http://ltblekinge.se/Forskning-och-utveckling/Blekinge-kompetenscentrum/Summary-in-English/.

## Supplemental Information

Supplemental information for this article can be found online at http://dx.doi.org/10.7717/peerj.1826#supplemental-information.

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
