# Peer review of "Questions of time and affect: a person’s affectivity profile, time perspective, and well-being"

_PeerJ, doi:10.7717/peerj.1826_

## Round 0.1 · original submission · Minor Revisions

· Academic Editor

Minor Revisions

Your manuscript received two reviews, both requesting revisions (see below). The process of re-review will be facilitated if you respond to the reviewers on an item-by-item basis and specify where changes have been made (yellow highlighting of revised text would be helpful). Be sure that reviewers' requests are addressed in the text, and not merely in the response to reviewers.

Reviewer 1 ·

Basic reporting

No Comments

Experimental design

No Comments

Validity of the findings

No Comments

Additional comments

Thank you for the opportunity to review this paper (“Questions of Time and Affect: A person’s affectivity profile, time perspective, and well-being”).

The authors examined differences between individuals in time perspective dimensions and examined the association between time perspective dimensions and well-being (which might be moderated by the individual’s type of Profile). This manuscript is a clearly written and well-argued piece of work. It addresses an important topic that will be of interest to readers of this journal. The literature cited appropriately included seminal works as well as recently published research. The methodology, analyses, and results were equally as well done.

However, there are some points that the authors should address:
1. Please explain briefly: Why was the Expectation-Maximization Algorithm used to replace missing values?
2. Results: Please avoid the term “effect” (line 331)
3. The limitations section must be extended: The problem of unobserved heterogeneity in subjective well-being research should be discussed.
4. Moreover, the sample (students/pupils, mean age 25.3±11.7) should be critically discussed.
5. Please note (minor): Zimbardo& Boyd, 1999; Garcia, Nima& Kyell, 2014, A valid reliableindividual-differencesmetric. Journal ofPersonality…);
6. Please use “” consistently (missing for the items of the Temporal Satisfaction with Life Scale).

·

Basic reporting

A few linguistic corrections are needed.
I have no other comment.

Experimental design

No comments.

Validity of the findings

An concise conclusion is warranted.
I have no other comments.

Additional comments

The authors present a study on the interactions of affective profiles and time perspective and their influence on psychological well-being. They investigated the effect of affective profiles and time perspective attitudes in 720 healthy individuals using multivariate analysis of variance (MANOVA) and structural equation modelling (SEM) and found different paths from time perspective to well-being between the affective profiles. The study is an original contribution to the body of literature that examines potential determinants of well-being, a topic of great importance in positive psychology. The design of the study was statistically sound and the manuscript is well-written. However, the paper would be strengthened by attention to the following issues:

1. In the Introduction: It would be helpful for the readers to give the description of the affective system immediately after the first statement with the terms ‘positive and negative affect’ (line 102). Moreover, a brief definition of ‘positive’ and ‘negative’ affectivity should be given here.
2. The authors should offer some information about other psychometric properties (criterion validity and test-retest reliability) of the Swedish version of the Positive Affect and Negative Affect Schedule, the Zimbardo Time Perspective Inventory, and the Temporal Satisfaction With Life Scale.
3. ‘Temporal life satisfaction was associated positively with past positive and negatively with past negative for all four profiles. Suggesting that these two time perspective dimensions are equally important independent of type of affective profile. However, at least for individuals with a self-destructive profile the future time perspective dimension was also associated to high levels of life satisfaction.’ Elaboration on these findings is needed.
4. The authors should offer a concise conclusion, in which they could also mention the implications of their findings for future research.

Minor comments

5. Some sentences should be corrected or rephrased:
- … to investigate differences between individuals in time perspective dimensions and to investigate if… (lines 33-35)
- The Swedish version has been used and in previous studies.. (lines 208-209)
- For example, regarding variances for the first MANOVA the ratio of largest (.27) to smallest (.10) variance was 2.70:1 (temporal satisfaction with life). For example, regarding variances for the second MANOVA… (line 301)
- … for each affective profiles. (line 314)
6. In line 352, write ‘(see Figure 1).’ instead of ‘See figure 1.’
7. Some of the references should be changed to follow the style of the journal.
8. The titles of Tables 1 and 2 and the legend of Figure 1 should be rephrased because they are syntactically incorrect.

---

## Round 0.2 · Minor Revisions

· Academic Editor

Minor Revisions

Thank you for all the corrections made in the manuscript. I am not convinced that the first question of the reviewer no 1 is fully answered about the use of the Expectation-Maximization Algorithm in order to replace missing values?

---

## Round 0.3 · accepted · Accept

· Academic Editor

Accept

I am pleased to inform you that your manuscript has been judged scientifically suitable for publication. Thank you for your contribution.